# Production of Low Molecular Weight Chitosan and Chitooligosaccharides (COS): A Review

**DOI:** 10.3390/polym13152466

**Published:** 2021-07-27

**Authors:** Cleidiane Gonçalves, Nelson Ferreira, Lúcia Lourenço

**Affiliations:** 1Institute of Technology, Graduate Program in Food Science and Technology, Federal University of Pará, Belém 66075-110, Pará, Brazil; luciahl@ufpa.br; 2Institute of Health and Animal Production, Amazon Rural Federal University, Belém 66077-830, Pará, Brazil

**Keywords:** hydrolysis, chitosan, molecular weight, chitooligosaccharides

## Abstract

Chitosan is a biopolymer with high added value, and its properties are related to its molecular weight. Thus, high molecular weight values provide low solubility of chitosan, presenting limitations in its use. Based on this, several studies have developed different hydrolysis methods to reduce the molecular weight of chitosan. Acid hydrolysis is still the most used method to obtain low molecular weight chitosan and chitooligosaccharides. However, the use of acids can generate environmental impacts. When different methods are combined, gamma radiation and microwave power intensity are the variables that most influence acid hydrolysis. Otherwise, in oxidative hydrolysis with hydrogen peroxide, a long time is the limiting factor. Thus, it was observed that the most efficient method is the association between the different hydrolysis methods mentioned. However, this alternative can increase the cost of the process. Enzymatic hydrolysis is the most studied method due to its environmental advantages and high specificity. However, hydrolysis time and process cost are factors that still limit industrial application. In addition, the enzymatic method has a limited association with other hydrolysis methods due to the sensitivity of the enzymes. Therefore, this article seeks to extensively review the variables that influence the main methods of hydrolysis: acid concentration, radiation intensity, potency, time, temperature, pH, and enzyme/substrate ratio, observing their influence on molecular weight, yield, and characteristic of the product.

## 1. Introduction

Chitin is the second most abundant polysaccharide in nature after cellulose. Chitin is found mainly in the exoskeleton of crustaceans and insects, in addition to bacteria, fungi, and mushrooms [1]. The partial deacetylation of chitin promotes the attainment of chitosan, and the difference between them is in the acetyl group. Chitin contains mainly units of N-acetyl-d-glucosamine (GlcNAc), while chitosan consists mainly of d-glucosamine (GlcN). The units that form both chitin and chitosan are linked by β (1 → 4) glycosidic bonds, Figure 1. In this context, it is understood that the greater the number of deacetylated units (GlcN) in chitosan, the greater the degree of deacetylation [1,2].

Chitosan has received attention as a functional biopolymer due to its cationic nature, biocompatibility, biodegradability, non-toxicity, and adsorption properties [2]. Its main characteristics are the molecular weight (Mw) and the degree of acetylation (DA) or degree of deacetylation (DDA), which correspond to the molar fractions of GlcNAc and GlcN. Most commercial chitosans have molecular weights ranging from 50–2000 kDa, with an average DDA of 50–100% (commonly 80–90%) [3]. Based on molecular weight, chitosan can be grouped into low molecular weight (<100 kDa), medium molecular weight (100–1000 kDa), and high molecular weight (>1000 kDa) [4].

The high molecular weight and the high degree of polymerization (DP) of chitosan result in low solubility at neutral pH. The high viscosity in solution is the main limitation in the food, cosmetics, agriculture, and health industry [5]. Therefore, to obtain chitosan with a more uniform molecular size and easy solubility, it is necessary to convert chitosan into oligomers. Chitosan with DP <20 and a molecular weight less than 3.9 kDa is called chitosan oligomers, chitooligomers, or chitooligosaccharides (COS) [3].

Chitooligosaccharides are the products of the hydrolysis of chitosan, and because they are soluble in water, they have several applications, such as antimicrobial, antioxidant, anti-tumor, and agricultural purposes [6,7,8,9,10,11,12].

Oligomers obtained from the hydrolysis of chitosan can be classified into homo-chitooligosaccharides, which are formed exclusively by GlcN or GlcNAc units, and also hetero-chitooligosaccharides. These latter are formed by units of GlcN and GlcNAc with varying degrees of deacetylation. Additionally, they can have different degrees of polymerization (number of monomer units within an oligomer) [13]. Regarding solubility, hetero-chitooligomers with DP <10 are considered soluble in water. However, water solubility with DP greater than 10 depends on the degree of deacetylation and the pH solution [14].

The way to obtain chitooligosaccharides can be by acid hydrolysis [4,15], oxidative [16,17], microwaves [18,19], gamma radiation [11,20], and enzymatic methods [21,22]. These hydrolysis methods provide chitosan with different molecular weights and degrees of deacetylation, which influences its composition, yield, and functionality. These methods have advantages and limitations, described in Figure 2.

Among hydrolysis techniques, acid hydrolysis has emerged as a convenient method for depolymerizing polysaccharides. However, acid hydrolysis generally requires severe treatment, as the rigid crystalline region in the chitosan granules inhibits acid penetration. For this reason, a high concentration of acid, ranging from 5 to 12 M, is used in most studies [9,23,24,25]. However, excessive acid loading can cause glucosamine degradation, which significantly reduces yield and generates major waste deposition problems [15,26]. In recent studies, a smaller amount of acid only made it possible to obtain medium molecular weight chitosan in a short hydrolysis time [4]. In high time, low molecular weight chitosan and chitooligosaccharides were obtained. However, yield decreased due to soluble products such as glucosamine monomers and dimers [15,27]. Therefore, current studies seek to develop the technique using different types and concentrations of acids associated with ionic liquids, induced electric fields, or other hydrolysis methods.

Oxidative hydrolysis, using hydrogen peroxide, is considered an easy and non-dangerous method. Because it is a slow method, most studies use this type of hydrolysis associated with other techniques [17,20]. In addition, to reduce the production cost, process variables are also studied [12,16].

The hydrolysis of chitosan by radiation has gained considerable attention because it is a relatively simple process, does not need to use a chemical reagent, is carried out at room temperature, and can be applied on a large scale [8,28]. This type of hydrolysis can have a high production cost if high doses of radiation are used. In this case, studies seek to use radiation associated with other reagents, especially hydrogen peroxide and acetic acid, to increase the efficiency of the process [11,29].

Otherwise, microwave hydrolysis is considered less efficient when compared to other high-energy radiation, such as ultraviolet and gamma. However, due to the lower environmental impact, microwave hydrolysis presents a viable alternative to obtain polysaccharides with molecular weights of interest [30]. In addition, due to the cost of the process, it is necessary to use less power and less hydrolysis time. For this, studies have associated this method with the use of hydrochloric acid, acetic acid, or ionic liquid [17,18,19]. However, it is necessary to be careful with the addition of these solvents in high concentrations, as they can generate residues and, consequently, a more significant environmental impact.

Enzymatic hydrolysis has received much attention because of its lower environmental risk. However, despite the enzymatic process taking place under mild conditions, the hydrolysis rate is slow in viscous solutions. Thus, a low substrate concentration (1% m/v) is used in most studies. The low substrate concentration causes an increase in the solution’s volume and the need for more enzymes. Additionally, specific enzymes have high prices and limited availability. Alternatively, low-cost enzymes from the hydrolases group can degrade chitosan [31,32,33].

Enzymatic hydrolysis is the only method that has association restrictions with other types of hydrolysis due to the sensitivity of the enzymes related to pH, high temperature, and radiation. Thus, most studies seek to optimize process variables (enzyme/substrate ratio, temperature, pH, and hydrolysis time) as an alternative to reduce costs [6,22,31,34]. Despite the disadvantages, enzymatic hydrolysis still seems to be the best alternative due to its selectivity. This selectivity favors the obtaining of chitosan chains with a specific average molecular weight [33].

Global interest in chitosan hydrolysis products has been steadily increasing. Based on the references cited in this review, the diversity of countries in which this theme has been highlighted was verified (Appendix A).

This review will present the main hydrolysis techniques of chitosan and analyze the main factors that influence the obtaining and characteristics of low molecular weight chitosan and chitooligosaccharides effectively at a lower cost.

## 2. Acid Hydrolysis

The acid hydrolysis of glycosidic bonds involves the following steps: (1) protonation of oxygen in the glycosidic bond through the connection of a proton (H_3_O^+^) to the glycosidic bond; (2) adding water to the final reducing sugar group; (3) decomposition of protonated glycosidic bonds (Figure 3). In this case, catalytic protons may be present in the water contained in the samples, and the protonated amino group of chitosan is also likely to act as a proton donor in catalysis [24].

### 2.1. Principal Acids for Hydrolysis

Acid hydrolysis is commonly performed by hydrochloric (HCl), acetic (CH_3_COOH), phosphoric (H_3_PO_4_), sulfuric (H_2_SO_4_), nitric (HNO_3_), and lactic (C_3_H_6_O_3_) acids (Table 1).

Among the acids studied, HCl is the most widely used due to its efficiency in the hydrolysis of the glycosidic bond (depolymerization) and the N-acetyl bond (deacetylation). HCl makes it possible to obtain fractions of the trimer, pentamer, hexamer, and heptamer-decamer. In this sense, chitooligosaccharides with DP ranging from 2 to 12 can be obtained from chitosan with a molecular weight of 658 kDa and 82% DDA, hydrolyzed with 6 M HCl at 70 °C, for 2 h [9].

In the study by Li et al. [15], there was a marked decrease in the molecular mass of chitosan (85.9% DDA) from 230 to 90 kDa in 12 h of hydrolysis with 1.5 M HCl using an induced electric field and temperature variation between 25 to 45 °C. In another study using chitosan of 200 kDa (90% DDA) hydrolyzed with HCl 2 M for 12 h, chitooligosaccharide of 2 kDa was obtained, with a yield of 85.2% [27].

In the kinetic study of acid hydrolysis, chitosan was quantitatively hydrolyzed to glucosamine in 6 h with 10 M hydrochloric acid (HCl) at 105 °C or 12 M HCl at 90 °C [25]. Despite being more used, the hydrolysis of chitosan with hydrochloric acid requires excess acid, complex reactors, and presents problems of deposition of residues. Excessive acid treatments result in the breakdown of glucosamine, which significantly reduces yield [26,37].

Thus, when chitosan is hydrolyzed in an HCl-H3PO4 solution, 75:25 in a molar ratio, a significantly higher efficiency (*p* < 0.05) is verified compared to HCl. This efficiency can decrease dramatically by increasing the proportion of H_3_PO_4_ [37]. However, when this comparison is made with other organic acids, hydrochloric and nitric acids require longer reaction times to obtain yields like those catalyzed with sulfuric acid [26].

### 2.2. Effect of Acid Concentration

In most studies, the types of acids and their concentrations are carefully established to hydrolyze chitosan (break glycosidic bonds) and minimize GlcN degradation (deacetylation). In this sense, the concentrations of sulfuric, hydrochloric, and nitric acids are directly proportional to the concentration of total reducing sugars, which are products of the hydrolysis of chitosan [26].

The concentration of the acid is also directly proportional to the rate of hydrolysis. Chitosan with a medium molecular weight (198.64 kDa) was obtained from hydrolysis with 5% acetic acid for 30 min. At concentrations of 1% and 3%, the molecular weight was 592.89 and 281.98 kDa, respectively [4]. In this case, it is observed that the lower acid concentration and short hydrolysis time enabled the production of only medium molecular weight chitosan.

In another study with HCl-H_3_PO_4_ acid solution (maintaining the molar ratio of 75:25) and 4 M [H+], the yield of 64.6% was observed due to incomplete hydrolysis of chitosan. Otherwise, 6 M [H+] showed an ideal recovery of 98.8%. However, by increasing the concentration to 9 M [H+], 90.3% was obtained due to the degradation of GlcN [37]. In this case, the increase in the acid concentration favored hydrolysis, but there is an upper limit to prevent GlcN degradation.

An alternative to decrease the high acid concentration is the association with other solvents, such as ionic liquids. The depolymerization of chitosan carried out by hydrochloric acid in the presence of ionic liquids shows a higher yield of total reducing sugars when compared to an aqueous system using only hydrochloric acid [26]. Additionally, chitosan is highly dissolved in ionic liquids, which facilitates the hydrolysis process [38,39].

### 2.3. Effect of Time

Studies have shown that increasing the hydrolysis time decreases the molecular weight, resulting in low molecular weight chitosan and chitooligosaccharides in the first hours. Even so, it is necessary to use a high concentration of acid. When using chitosan with a molecular weight of 214 kDa (96% DDA), hydrolyzed in 85% phosphoric acid solution (60 °C), a high decrease was obtained in the first 4.0 h (74 kDa), and, after 15 h, molecular weight reached 19 kDa [36]. In another study with a 2 M HCl solution, the molecular weight of chitosan was reduced from 200 to 29, 17, and 3.0 kDa in 30 min, 1.0 h, and 5.0 h, respectively [27]. In chitosan 2038 kDa, hydrolyzed in 5 M HCl at 65 °C, the molecular weight after 5.0 h was 156 kDa, reaching 74 kDa in 36 h [24].

In addition, the yield of the hydrolyzate product may decrease when the reaction time is prolonged. In this case, more chitosan is hydrolyzed to soluble products such as glucosamine monomers and glucosamine dimers. When phosphoric acid at 85% is used, the yield in 1 and 15 h is 89.3% and 49.2%, respectively [36]. In a 2 M hydrochloric acid solution, the yield is 92.1% and 80.1% over 0.5 and 24 h, respectively [27]. A longer hydrolysis time in HCl-H_3_PO_4_ solution (4.5:1.5 M) will be necessary to combine the two acids. However, it shows an increase in the relative yield with the hydrolysis time up to 24 h. After that time, glucosamine degrades slightly, decreasing from 97.7% in 24 h to 90.0% in 36 h. In addition to the decrease in yield, the solution may darken with the hydrolysis time for more than 24 h [37].

### 2.4. Effect of Temperature

The hydrolysis temperature also influences the molecular weight and yield of chitosan. Thus, chitosan with 214 kDa and hydrolyzed in a phosphoric acid solution (85%) at temperatures of 40, 60, and 80 °C for 8.0 h produced low molecular weight chitosan of 37, 35, and 20 kDa, respectively. This shows that the higher the temperature, the lower the molecular weight of chitosan. However, the yield will be lower. In this case, the yield was 86%, 71%, and 61%, respectively [36].

Chitosan of initial molecular weight of 726 kDa (medium molecular weight) and 28 kDa (low molecular weight) hydrolyzed in lactic acid (1%) at temperatures of 8, 22, and 37 °C also showed an increase in the rate of hydrolysis with the increasing temperature. However, there was an influence of the initial molecular weight of chitosan on the hydrolysis rate in this case. This rate increased from 25% to 50% with the increase in temperature in medium molecular weight chitosan. The effect of temperature was less on low molecular weight chitosan. In these conditions, rates of 5% to 10% were observed [35]. 

When the three most essential variables in acid hydrolysis (acid concentration, time, and temperature of hydrolysis) were evaluated in the same process, the yield of glucosamine reached 98% in the condition of 10 M HCl and 105 °C for 6 h, a recovery also being observed such as using 12 M HCl and 90 °C for 6 h. In this case, it was observed that the acid concentration had a better influence on the reaction [25]. However, when analyzing only the effect of time and temperature of hydrolysis in acetic acid solution, it was observed that these variables have the same influence [4].

### 2.5. Influence of the Degree of Deacetylation and Initial Molecular Weight of Chitosan

The degree of deacetylation is inversely proportional to the rate of hydrolysis. It was observed that chitosan with a deacetylation degree of 85%, 67%, and 53% showed a decrease in viscosity of 40%, 50%, and 60%, respectively, after 10 days of hydrolysis in 1% lactic acid [35]. This behavior was confirmed in a study with chitosan with an initial molecular weight of 230 kDa and a degree of deacetylation of 86% and 55%. It was observed that after 12 h of hydrolysis, there was a decrease in molecular weight to 90 and 30 kDa, respectively [15].

The initial molecular weight influences the viscosity reduction rate. Chitosan hydrolyzed in 1% lactic acid with medium molecular weight (726 kDa) showed a 90% decrease in viscosity in 180 days. This decrease is most evident during the first 10 days. In low molecular weight chitosan (28 kDa), the viscosity decreases by only 51% in 180 days [35].

### 2.6. Antimicrobial and Antioxidant Properties of Chitosan Oligomers

The hydrolysis product of chitosan under conditions (HCl 6.7 M; 56 °C; 3 h) had an effect against Gram-positive bacteria, *Staphylococcus aureus* ATCC 25923, *S. aureus* ATCC 43300, *Bacillus subtilis,* and *Bacillus cereus,* and Gram-negative bacteria, *Escherichia coli* ATCC 25922, *Pseudomonas aeruginosa* ATCC 27853, *Salmonella typhimurium*, *Vibrio cholerae*, *Shigella dysenteriae*, *Prevotella melaninogenica,* and *Bacteroides fragilis* [23].

Another characteristic of the product of chitosan hydrolysis is the antioxidant activity. Fractions with different degrees of polymerization were obtained by hydrolysis with 6M HCl. In this sense, trimer, tetramer-pentamer, pentamer, hexameter, and heptamer-decamer showed activity against hydroxyl radicals. The activity was greater with the decrease in the degree of polymerization due to the chito-oligomers having shorter chains and weak intramolecular and intermolecular hydrogen bonds. This fact allows the hydroxyl and free amino groups to be activated, which helps to promote antioxidant activity [9].

## 3. Gamma Radiation Hydrolysis

The hydrolysis of chitosan by gamma radiation (γ) has gained considerable attention due to its advantages. Among these advantages, there is the possibility of carrying out the process at room temperature and applying it on a large scale [40]. This type of hydrolysis is mainly influenced by the intensity of the radiation dose and the hydrolysis solution (Table 2).

The radiation causes changes in the polymer chain structure, inducing the formation of radicals and breaking the chain. The mechanism of hydrolysis of chitosan by radiation is described by Kim et al. [44]:$$R\text{–}H\;\overset{hv}{\to }\;H·\;(C4_{\;}\text{–}C_{6})+H$$
R–H+ H· → R· (C_1_–C_6_) + H_2_
R· (C_1_–C_6_) → F_1_· + F_2_ (Scission)
R–NH_2_ + H· → R· (C_2_) + NH_3_
where R–N and R–NH_2_ are chitosan macromolecules, and R· (Cn) is a chitosan macroradical located on a Cn carbon atom. F_1_· and F_2_ are fragments of the main chain after scission.

### 3.1. Effect of Radiation Intensity

The increase in radiation intensity will directly influence the decrease in the molecular weight of chitosan. Although intensity is an important parameter, there is a limit on the radiation dose to make the process effective.

When 0.5–200 kGy radiation was used, 50 kGy was needed to obtain the chitooligosaccharides with a decrease in molecular weight from 67 to 0.97 kDa [43]. In another study using chitosan irradiated between 2–200 kGy, there was a decrease in viscosity with up to 10 kGy. In this case, the increase in radiation only increased the monomer, dimer, and trimer amounts [41]. In chitosan irradiated with 20–200 kGy, the molecular weight decrease occurred rapidly up to 120 kGy. Subsequently, the rate of degradation was low [7].

In the study by Dung et al. [29], the chitosan of 193 kDa, 80% DDA, showed a decrease in molecular weight to 11.4 kDa with irradiation up to 75 kGy and then slowly decreased with an increase in irradiation up to 150 kGy. Using chitosan 210 kDa after hydrolysis with radiation of 2, 10.0, and 20 kGy, low molecular weight chitosan and chitooligosaccharide of 35, 6.0, and 2.0 kDa, respectively, were obtained [28].

The variation in the radiation dose required to hydrolyze chitosan is due, among other factors, to the initial molecular weight of chitosan. In chitosan with an initial molecular weight of 338 kDa, 82% DDA, irradiated at 100 kGy, there was a significant decline in molecular weight to 82 kDa [11]. In Choi et al. [41], chitosans with molecular weights of 61 and 110 kDa showed, after hydrolysis with irradiation of 10 kGy, a decrease in viscosity to 99.8% and 95.7%, respectively. These studies show that the initial molecular weight of chitosan is directly proportional to the radiation dose required for hydrolysis.

### 3.2. Hydrolysis Solution

The intensity of radiation is also influenced by the environment/solution of chitosan during hydrolysis. In aqueous solutions, the influence occurs mainly by water radiolysis due to the formation of transient products that react with the solute. The subsequent reaction of the macroradicals can be chain scission, hydrogen transfer, inter-and intramolecular recombination, and disproportionation macroradicals [43]. These reactions lead to the creation of hydroxyl radicals, which can bind strongly to the β-1-4 glycosidic bonds, favoring hydrolysis [44].

Hydrolysis of chitosan by radiation can occur under different conditions. These conditions can be identified as dry, moist, and in solution (acetic acid/H_2_O_2_). Chitosan with 220 kDa, 73% DDA, in dry and moist form showed a reduction in molecular weight to 180 and 160 kDa with 15 kGy, respectively. However, from 25 kGy onwards, a similar trend was observed in the reduction of molecular weight. This process was more efficient in the moist form due to the radiolysis of the absorbed water that initiated the fission reactions in the polymer chain. More efficiently, chitosan in H_2_O_2_ solution showed a significant reduction in molecular weight to 40 kDa, even at lower doses such as 15 kGy. This hydrolysis is enhanced due to the combined action of water radiolysis and H_2_O_2_, reaching 10 kDa in 150 kGy against 50 kDa of chitosan in the wet form [8].

The same trend was obtained when using chitosan with a molecular weight of 90 kDa, 70% DDA, irradiated in the presence of H_2_O_2_ (1%). The irradiation showed a highly effective result even in low doses compared to previous studies, with a decrease in molecular weight to 8.6 and 2.7 kDa, with 4 and 16 kGy, respectively. The results indicate that the molecular weight obtained after hydrolysis was much lower when there was a combination of radiation with H_2_O_2_ [40]. In another study with chitosan 173 kDa, 96.9% DDA, it was also evident that lower radiation of 6 kGy associated with hydrogen peroxide 1%, 3%, and 5% reduced the molecular weight to 14, 11, and 8 kDa, respectively. Less efficient results of hydrolysis were observed without the addition of H_2_O_2_ [20].

Chitosan with 400 kDa and 90.5% DDA showed a higher rate of hydrolysis with the combination of gamma radiation (50 kGy) and hydrogen peroxide (2%). The reduction in molecular weight reached 6 kDa and was more efficient when using only radiation. In this case, the molecular weight of the hydrolyzed chitosan was 100 kDa [42].

When the chitosan was hydrolyzed by radiation in the presence of H_2_O_2_, hydroxyl radicals formed due to the water radiolysis. Hydroxyl radicals are efficient oxidants that react with chitosan to abstract hydrogen linked to carbon. Subsequently, the resulting carbohydrate radicals cause the breakdown of glycosidic bonds by rearrangement, which reduces the molecular weight of chitosan very effectively [29,42].

The use of other initiators can also increase the rate of hydrolysis. Chitosan with 10^7^ Da molecular weight, irradiated at 80 kGy, showed a higher rate of hydrolysis in the presence of initiators such as potassium persulfate and ammonium persulfate when compared only with ionizing radiation. In this context, ammonium persulfate was the most effective initiator in hydrolysis in chitosan. It was also observed that the use of ammonium persulfate with 40 kGy decreased the molecular weight by 98%, while the sample hydrolyzed by radiation had a reduction of 25% [7].

Based on these results, it can be inferred that the presence of initiators is an alternative to decrease the radiation dose required to hydrolyze chitosan. As previously mentioned, this dose may be higher with the increase in the initial molecular weight of chitosan. From an economic point of view, high doses of radiation may not be accepted due to the high cost.

### 3.3. Macroscopic Changes

In addition to the change in molecular weight due to the cleavage of the β-(1-4) glycosidic bond, a positive characteristic of radiation hydrolysis is that the irradiated chitosan maintains its degree of deacetylation, regardless of the radiation load. A study based on chitosan with an initial deacetylation degree of 91.5% shows that deacetylation was 91% and 91.2% at doses of 10 and 20 kGy, respectively [28].

Another characteristic of irradiated chitosan is the change in color from creamy yellow to dark brown when the radiation dose is above 100 kGy. The color change can occur because of the oxidation of chitosan at the molecular level and C = O due to the chain fission reaction [11,41].

### 3.4. Antioxidant and Antimicrobial Properties of Chitosan Oligomers

The radiation intensity directly influences the antioxidant activity of chitosan due to the change in molecular weight, because the lower the molecular weight, the greater the antioxidant activity [11]. In a study with different radiation doses between 2 and 20 kGy, the percentage of elimination of hydroxyl radical from 41% to 64% was observed. Additionally, at 20 kGy, the elimination of 74% of the superoxide radical was observed. From these results, radiation at 20 kGy was observed, providing sufficient hydrolysis to increase chitosan activity due to the decrease in molecular weight. In this sense, chitosan with high molecular weight has a compact structure. This compacted structure makes intramolecular hydrogen bonds more effective. Consequently, there will be a decrease in the hydroxyl and amino groups [28].

The microbial activities against Gram-positive and Gram-negative bacteria are also influenced by the molecular weight and, consequently, the radiation intensity. Thus, chitosan with medium and low molecular weight of 220, 120, 75, and 52 kDa have high antibacterial activity with less molecular weight and a higher radiation dose. Positive activity against *Staphylococcus* sp, *Pseudomonas aerogonisa*, *Escherichia coli*, and *Staphylococcus aureus* was evaluated. The main reason may be related to chitosan with lower molecular weight to access the microbial cell more easily and alter cellular metabolism [8].

## 4. Microwave Hydrolysis

The use of microwave irradiation to accelerate organic reactions has increased interest due to being environmentally positive. This hydrolysis mechanism is a complex process and probably involves two separate paths, such as shear forces induced mechanically by the oscillation of molecules (breaking the main polymer chain) and thermal degradation (heat-induced hydrolysis) [30]. Chito-oligosaccharides formed mainly by disaccharides, trisaccharides, tetrasaccharides, pentasaccharides, and less hexasaccharide can be obtained [17]. This type of hydrolysis is mainly influenced by the power used, solution, and hydrolysis time (Table 3).

### 4.1. Microwaves versus Conventional Heating

The use of microwave irradiation can reduce the reaction time compared to the conventional method of heating in a thermostatic bath or shaker incubator. Chitosan with an initial molecular weight of 560 kDa, 85% DDA, 25 min, showed a higher hydrolysis rate (105 kDa) with microwaves when compared to conventional heating (270 kDa) at the same temperature (100 °C) for 2 h [47].

In another study with chitosan 66 kDa, 95% DDA, hydrolyzed with 100 W for 20 min and equilibrium temperature of 89 ± 2 °C, chitosan of 26 kDa was obtained. Otherwise, in conventional hydrolysis, in a water bath, at 89 °C, for 20 min, the hydrolysis was significantly lower (54 kDa). These aspects led to the conclusion that thermal effects are not the only degradation mechanisms involved. The most likely simultaneous process is mechanical shear induced by molecular vibrations [30].

In addition to the reaction time, the product yield is higher in microwave hydrolysis. In the study with microwave irradiation (800 W) in chitosan of 560 kDa, 90% DDA, at 155 °C for 2 min, a yield of reducing sugars of 87% was observed, while in conventional heating, in an oil bath at 150 °C, was 38.9% for 5 h [18].

### 4.2. Hydrolysis Solution

The hydrolysis of chitosan through microwave irradiation can be accelerated by adding inorganic salts. Thus, chitosan with a molecular weight of 5.6 × 10^5^ Da decreased to 10^4^ (without adding salt) and 3 × 10^4^ Da (with adding salt). This is due to the salt’s ability to cause the solution to overheat. Microwave heat involves direct interaction with certain classes of absorbent molecules that can lead to the introduction of energy and raise the temperature of the solution. Additionally, the addition of salts to solvents can increase conductivity and influence the rate of heating. The presence of salts in polar solvents also improves dielectric loss and the microwave coupling of the solvents [47].

Another way to accelerate the rate of hydrolysis is through the hydrogen peroxide associated with the microwave. In this case, the concentration of hydrogen peroxide is proportional to hydrolysis, obtaining a recovery rate of chitosan dissolved in water of 52% and 92% when using 10% and 15% of this reagent, respectively. Additionally, chitosan with a molecular weight of 220 kDa, 92% DDA, hydrolyzed in a microwave (700 W), in a 15% hydrogen peroxide solution for 4 min, had a molecular weight of 0.9 kDa [45]. When using 658 kDa chitosan, hydrolyzed with 800 W for 25 min, in a 1% hydrogen peroxide solution, chitooligosaccharide with a molecular weight of 1.46 kDa was obtained [17].

Most studies of hydrolysis of chitosan through microwave irradiation use acids to dissolve chitosan. These solvents leave an environmental liability and promote the corrosion of the equipment. However, the hydrolysis of chitosan can be carried out using “green solvents”. An alternative is to use ionic liquids as suitable solvents to hydrolyze polysaccharides. Thus, when the chitosan hydrolysis is carried out with microwave irradiation (640 W), using ionic liquids, it is observed that the molecular weight of the chitosan decreases from 560 to 0.88 kDa in 90 s. In this case, the yield of reducing sugars was 84% and can reach 92% in 2 min. Additionally, to reduce the viscosity of the reaction system, dimethyl sulfoxide can be added as a co-solvent, forming a homogeneous solution. Polar materials in a microwave field can quickly reach high temperatures and a higher reaction rate [18].

### 4.3. Effect of Time

The reaction time also influences the molecular weight of microwave-hydrolyzed chitosan. The hydrolyzed chitosan with 100 W (without additives and initiators) showed a significant decrease in molecular weight from 66 to 42, 34, and 26 kDa in times of 5, 10, and 20 min, respectively. After this time, there was no change in molecular weight [30].

The increase in hydrolysis time with the inorganic salt also favors the decrease in viscosity and consequently in molecular weight [47]. However, the rate of hydrolysis in inorganic salt is low compared to ionic liquids. Chitosan of 560 kDa, 90% DDA, hydrolyzed with microwave (640 W) and ionic liquids showed a decrease in molecular weight, reaching 24, 0.88, and 0.45 kDa for 30, 90, and 120 s, respectively. This indicates that the combination of microwave irradiation in the presence of ionic liquids attacks 1,4-β-glycosidic bonds very efficiently [18].

### 4.4. Power during Hydrolysis

Another variable that influences the efficiency in breaking glycosidic bonds is the potency during hydrolysis. In chitosan with a molecular weight of 175 kDa, 83% DDA, in the power of 390 W (30 min) and 650 W (10 min), low molecular weight chitosan of 103 and 79 kDa were obtained, respectively [46]. In other studies, with 80–100 [19] and 10–100 W, the same behavior was also observed because a higher power has a higher shear force between molecules, causing a faster decrease in molecular weight [30].

This behavior was also observed in the study with powers of 160, 320, 480, 640, and 800 W, where it was verified that the final temperature reached 120, 125, 130, 146, and 155 °C, respectively, and the corresponding reducing sugar yields were 35%, 67%, 74%, 93%, and 87%. In this case, the highest power, 800 W, showed a lower sugar yield than the 600 W power attributed to the decomposition of the product, caused by high temperature [18].

## 5. Oxidative Hydrolysis with Hydrogen Peroxide

Hydrogen peroxide (H_2_O_2_) is used for the hydrolysis of polysaccharides because it is easy to handle, readily available, and environmentally friendly [16]. This method is based on the formation of reactive hydroxyl radicals by dissociating hydrogen peroxide. Hydroxyl radicals, which are robust oxidizing species, can attack β-d- glycosidic bonds (1 → 4), resulting in the hydrolysis of chitosan [48].

The depolymerization of chitosan by hydrogen peroxide provides a breakdown of the 1,4-β-d-glucoside bonds of the polysaccharide chain, leading to a decrease in molecular weight [49]. Chito-oligosaccharides obtained by oxidative hydrolysis are mainly composed of monosaccharides to pentasaccharides. The most significant amount found is disaccharides and trisaccharides, with or without acetylation [17]. Tian et al. [49] present the hydrolysis mechanism described below:R–NH_2_ + H^+^ ↔ R–NH_3_^+^
H_2_O_2_ ↔ H^+^ + HOO^−^
H_2_O_2_ + R–NH_2_ + H^+^ ↔ R–NH_3_^+^+ HOO^−^ + H^+^

The hydroperoxide anion is very unstable and is easily decomposed into a highly reactive hydroxyl radical (HO·).
HOO^−^ → OH^−^ + O
H_2_O_2_ + HOO^−^ → HO· + O_2_·^−^ + H_2_O

Eventually, the hydroxyl radical (OH·) attacks the glycosidic bond of chitosan to produce the chitosan oligomer according to the reactions below:(GlcN)_m_–(GlcN)_n_ + HO· → (GlcN·)_m_–(GlcN)_n_ + H_2_O
(GlcN·)_m_–(GlcN)_n_ + H_2_O → (GlcN)_m_ + (GlcN)_n_

During hydrolysis, R-NH_2_ reacts preferentially with H^+^ to produce R-NH_3_^+^, which causes a decrease in H^+^ and an increase in pH. Additionally, HOO^−^ is rapidly decomposed into HO ·, which means that H_2_O_2_ is continuously decomposed. These radicals undergo other reactions quickly to form low molecular weight water-soluble oxidation products.

This type of hydrolysis is mainly influenced by the concentration of hydrogen peroxide, time, and temperature of hydrolysis, Table 4.

### 5.1. Concentration of Hydrogen Peroxide

The concentration of hydrogen peroxide is directly proportional to the rate of hydrolysis. However, its high concentration can influence the elimination of hydroxyl radicals and reduce the reaction efficiency. In this sense, the concentration of 2% H_2_O_2_ is ideal for chitosan hydrolysis [48].

In addition to the hydrolysis rate, the yield is also maximum (62%) when the H_2_O_2_ concentration reaches 2%. From this concentration, the yield decreases to 58% and 55% at concentrations of 2.5% and 3%, respectively. This decrease can be attributed to the oligosaccharides production with a shallow degree of polymerization that makes ethanol precipitation difficult [12].

The concentration of reducing sugars is also maximum (14.8%) when using 2% H_2_O_2_. However, from the concentration of 2.5% and 3%, it was observed that the amount of reducing sugars reduced to 14% and 12%. This occurred due to the aldehyde oxidation when there is an excess of H_2_O_2_ [16].

### 5.2. Association with Other Types of Hydrolysis

The formation of radical groups is practically inefficient when hydrogen peroxide is used alone. For this reason, to improve the efficiency of hydrolysis, several studies associate hydrogen peroxide with other patterns of degradation. Among them are some works previously discussed using gamma radiation [7,8,20,40,42] and microwaves [17,45].

In addition to these methods, chitosan is also effectively hydrolyzed by hydrogen peroxide under ultraviolet irradiation. In this case, when chitosan is hydrolyzed only in the presence of hydrogen peroxide, viscosity decreases by 20% and 63% at 30 and 180 min, respectively. However, when hydrogen peroxide is combined with ultraviolet irradiation, viscosity decreases by 84% and 92% for the same analyzed times [48].

### 5.3. Association with Other Reagents

Another way to increase the efficiency of the process is to hydrolyze chitosan using hydrogen peroxide under the catalysis of phosphotungstic acid. Phosphotungstic acid is a heteropolytic acid that presents simple preparation and high reactivity and is non-corrosive, in addition to having acid resistance and relatively high thermal stability [16,50]. The rate of hydrolysis of chitosan at 70 °C for 30 min without the catalyst was only 43%. This indicates that degradation is inefficient when H_2_O_2_ is used alone. However, when 0.1% phosphotungstic acid was used, the rate was 99.32% [50]. 

The concentration of phosphotungstic acid influences the hydrolysis product. There is a sharp increase in the concentration of reducing sugars with a higher concentration of phosphotungstic acid from 0.04% to 0.1%, without an additional increase from 0.1% [16].

When associated with acetic acid, its molar ratios influence the recovery of chitooligosaccharides. Therefore, the higher the concentration of hydrogen peroxide about acetic acid, the higher the yield of chitooligosaccharides (dimers–decamers), with a yield of 36%, 22%, 18%, and 14%, for molar ratios of 5.7, 2.8, 1.9, and 1.7 respectively, in addition to obtaining chitosan with lower molecular weight, 6.61, 7.7, 9.29, and 9.03 kDa, respectively [46]. 

### 5.4. Effect of Time

As with other types of hydrolysis, time will also influence when using hydrogen peroxide. In chitosan hydrolyzed by H_2_O_2_ in HCl solution (0.9%) for 0.5–8 h, there was a reduction in molecular weight with the increase in hydrolysis time. A rapid decrease from 480 to 50 kDa in 0.5 h has been shown, with a 95% yield [51]. In the study using hydrogen peroxide and acetic acid, the yields of chitooligosaccharides increased considerably with increasing hydrolysis time, reaching an optimal yield of 62.42% in 4 h [12].

The reaction rate is higher when H_2_O_2_ hydrolyzes the chitosan under the catalytic action of phosphotungstic acid with an ideal time of 30 min of hydrolysis. In these conditions, a higher concentration of reducing sugars was obtained in 30 min and no further increase after 40 min [16]. Additionally, there is a reduction in the molecular weight of chitosan from 700 to 4.7 and 4.3 kDa in 30 and 120 min, respectively. This shows that after 30 min, under the catalysis of phosphotungstic acid (0.1%), there will be no significant reduction in molecular weight [50].

### 5.5. Effect of Temperature

Temperature is another variable that influences the reaction, since the temperature is directly proportional to the hydrolysis rate. This behavior was observed in 498 kDa chitosan, hydrolyzed in H_2_O_2_ solution (0.3%) for 2 h at 30, 50, and 90 °C. The obtainment of medium and low molecular weight chitosan of 200, 25 Da, and 5 kDa, respectively, was observed [51].

Additionally, the yield of chitosan after H_2_O_2_ hydrolysis is proportional to the increase in temperature, with a maximum concentration of reducing sugars at 65 °C. However, when the reaction temperature exceeds 65 °C, browning occurs in the reaction mixture and a decrease in the concentration of reducing sugars due to oxidation of the aldehyde. Therefore, the most suitable maximum reaction temperature is 60 °C [12,16,49].

## 6. Enzymatic Hydrolysis

Hydrolysis of chitosan catalyzed by enzymes is more specific compared to other types of hydrolysis. The specificity allows better control of the extent of the reaction and the size of the oligomer. Thus, enzymatic hydrolysis tends to be an attractive alternative [32]. Enzymes act specifically in reactive sites, either internally fragmenting the molecule or acting from one end, releasing monomers or dimers sequentially [31].

Chitosanases are the specific enzymes intended for the hydrolysis of chitosan. However, these enzymes have reduced commercial availability and, consequently, have high commercial value, presenting limited industrial use [32]. Therefore, as a lower-cost alternative, other commercially available enzymes capable of hydrolyzing polysaccharides in large quantities are studied (Table 5). These discoveries enable the development of efficient and economically viable industrial processes for the hydrolysis of chitosan.

### 6.1. Hydrolysis of Chitosan by Different Enzymes

Different commercial enzymes were tested to decrease the viscosity of chitosan solutions and release the reducing ends of the polymer. Among these enzymes, cellulase, pepsin, and lipase proved to be more suitable for the hydrolysis of chitosan at a level comparable to that obtained by chitosanase, with a decrease in viscosity in 69%, 80%, 82%, and 65%, respectively. It was observed that some hydrolases might also have the ability to act on the reducing ends of chitosan in the first hour of hydrolysis. This is possible because of the cleavage specificities of exohydrolases [32].

In addition to the hydrolytic activity exo-1,4-β-d-glucanases, which cleaves glycosidic bonds present in the reducing ends of polymers, hydrolysis can also occur in endo-1,4-β-glucanases, which cuts off the β- 1,4-internal glycosides [60]. The performance of the endogluganase enzyme studied by Roncal et al. [32] could be observed through kinetics about viscosity. A decrease in viscosity was observed in the first 20 min. Thus, the greatest reduction in viscosity is better observed in the large chitosan chains than in the smaller chains.

The exact amount of unit of activity (100 U/mL) for each enzyme, in the hydrolysis of chitosan (371.5 kDa), presented a more efficient catalyst capacity in 96 h (<6 kDa) than chitinase (11.2 kDa) and lysozyme (22.2 kDa) [53]. Thus, due to the high rate of cellulase hydrolysis, in addition to greater availability and lower cost, most studies are increasingly focusing on the use of cellulases for the hydrolysis of chitosan.

The species that produce cellulases can influence hydrolysis rates. The purified cellulases obtained from *Trichoderma viride*, *Trichoderma reesei*, and unpurified cellulase obtained from *Aspergillus niger* have hydrolytic activity. However, *A. niger* and *T. reesei* may have high hydrolysis rates, while *T. viride* cellulase may be lower. In this case, the cellulase efficiency of the Aspergillus genus is possibly due to its better β-glycosidase activity compared to the Trichoderma genus [33].

Aspergillus cellulase has a good capacity for enzymatic hydrolysis, being a viable alternative to replace chitinase and chitosanase, which are enzymes of more excellent economic value [33,59]. Cellulase can hydrolyze chitosan 84% DDA and molecular weight from 518 to 35 kDa in the first hour of hydrolysis at 50 °C [33].

Although cellulase has a non-specific hydrolytic action on chitosan, it can be observed in previous studies that this enzyme has excellent hydrolysis capacity. This is due to the structural similarity of chitosan with cellulose (Figure 4). Cellulose is formed by d-glucose polymers joined by β-1,4-glycosidic bonds. In chitosan, the C-2 hydroxyl groups are replaced by amino groups, and apparently, the enzyme does not efficiently recognize the C2 position group in the glucose or glucosamine residues. For the same reasons, chitosanase and cellulases also exhibit high homogeneity, and it is not uncommon for them to be found in the same microorganism [60].

### 6.2. Products of Enzymatic Hydrolysis

The structures of the chitooligosaccharides formed also depend on the specificity of the enzymes used in the enzymatic pool. In this sense, the enzymatic hydrolysis of chitosan at 39 °C, pH 5.3, for 24 h, from the mixture of pectinase from *Rhizopus oryzae*, cellulase from *Aspergillus niger* and papain, resulted in chitosan fractions with a molecular weight of 34 and 14.6 kDa. The content of chitooligosaccharides obtained was 33% of dimers–octamers and 54% of dimers–tetramers [46]. When only the *Aspergillus niger* cellulase enzyme was used at 50 °C, pH 5.6, for 24 h, the products were formed mainly by chitooligosaccharides with a degree of polymerization from 3 to 11 [59]. Otherwise, the use of papain alone resulted in a mixture of oligosaccharides and monomers (GlcN and GlcNAc), with four main components GlcNAc, GlcN, (GlcN)3, and (GlcN)4 [55].

Glucosamine (GlcN) and N-acetyl-glucosamine (GlcNAc) oligomers were identified as products of the depolymerization of chitosan when commercial cellulase was used. This result indicates that the enzyme cleaved the GlcN-GlcN and GlcNAc-GlcN bonds. Thus, from hydrolysis, 60 °C, pH 5.2, for 0.5–12 h, chitobiose (GlcN)_2_, chitotriose (GlcN)_3_, chitotetraose (GlcN)_4_, and some chitooligosaccharides with long-chain length were obtained. Shorter d-glucosamine oligomers with increased hydrolysis time have also been produced [54].

An enzymatic solution of *Bacillus amyloliquefaciens* was used to obtain oligomers through the hydrolysis of chitosan at 37 °C at pH 5. The quantities of oligomers (GlcNAc)n, *n* = 1–6 increased with the hydrolysis time between 1 and 12 h. After hydrolysis for 12 h, the quantities of all oligomers increased, but (GlcNAc)6 was the lowest among the oligomers (GlcNAc)n, *n* = 1–6 in the composition of the hydrolysates. After hydrolysis for 24 h, the hydrolysates composition was almost all (GlcNAc) n, *n* = 1–3. The amount of monomer that is the most found in the composition of the hydrolysates was GlcNAc [10].

Some bacteria also produce enzymes of interest for the hydrolysis of chitosan. The enzymatic extract obtained from a commercial preparation of *Bacillus thuringiensis* efficiently hydrolyzed chitosans with a deacetylation degree of 81% and 90% and showed a higher reaction rate for the more deacetylated chitosan. This made it possible to obtain partially acetylated GlcNAc-(GlcN)1-3 and deacetylated (GlcN)2-5 oligosaccharides. Chitosan with 90% deacetylation was converted to oligosaccharides in 55 h. Most products were 16% chitobiose, 17% chitotriose, 50% chitotetraose, and 14% chitopentaose [57].

The hydrolysis products resulting from the action of cellulase, pepsin, lipase, and chitosanase on chitosan were divided into two fractions. The first fraction is insoluble (low molecular weight chitosan), with a yield of 49%, 48%, 50%, and 45%, respectively. The second is the soluble fraction, formed by oligosaccharides and monomers. Oligosaccharides had yields of 46%, 52%, 42%, and 46%, respectively. The yield of the GlcN and GlcNAc monomers was not identified for pepsin. However, for the other enzymes, the yield was 5%, 7%, and 9%, respectively. This particular property of pepsin allowed to produce considerably more oligosaccharides than other enzymes. In this case, we can infer that pepsin is an excellent alternative for obtaining oligosaccharides due to the rapid decrease in viscosity from short and medium-chain oligosaccharides without monomer generation [32].

A new chitosanase (GsCsn46A) from *Gynuella sunshinyiii* showed high hydrolytic activity when producing oligosaccharides in a moderate reaction condition (pH 5.5) at 30 °C/30 min, where it was obtained mainly (GlcN)_2_ to (GlcN)_7_. After 2 h of incubation (GlcN)_2_, (GlcN)_3_, (GlcN)_4_, and a smaller amount of (GlcN)_5_ were obtained. However, only (GlcN)_2_ and (GlcN)_3_ were detected after 6 h of reaction. The yield rates of chitooligosaccharides ranged from 71% to 94% from 30 min to 6 h [56].

### 6.3. Effect of Temperature on Enzymatic Hydrolysis of Chitosan

The temperature on enzyme activity is related to two established thermal parameters. The first is the Arrhenius activation energy, which relates to the effect of temperature on the catalytic reaction. The second is thermal stability, which shows the effect of temperature on thermal inactivation [62].

The ideal value in the hydrolysis rate depends on the type of enzyme. In the study using the enzymatic complex Celuzyme^®^ XB, formed by cellulase, xylanase, and β-glucanase, the highest hydrolysis rate was reached at 51 °C, reaching a relative activity of 90%. Additionally, the increase in temperature tends to decrease its catalytic activity. This occurs due to the modification of the tertiary structure that may lose its active and functional conformation [31]. This behavior was also observed when using pepsin, which showed an increase in hydrolysis activity by increasing the temperature to 45 °C, with a relative enzymatic activity of 75% [32].

Cellulase acted significantly at 50 °C with a decrease in viscosity above 90% in 15 min. However, hydrolysis was lower at 60 °C, where the reduction was 20%. In this case, cellulase was relatively stable at a temperature below 55 °C and was quickly inactivated at higher temperatures [33].

### 6.4. Effect of Time on Enzymatic Hydrolysis of Chitosan

The molecular weight of chitosan decreases with increasing hydrolysis time. However, the intensity of the effect may vary according to the synergistic effect of different enzymes. Thus, enzymes that exhibit endo hydrolysis characteristics may present a higher rate of hydrolysis at the beginning of the reaction. Chitosan hydrolyzed by papain, cellulase, and lysozyme showed a rapid decrease in molecular weight in the first hour of hydrolysis, with a reduction of 194 kDa (90% DDA) to 41, 80, and 150 kDa, respectively. After 8 h of hydrolysis, the values were 5, 12, and 16 kDa, respectively. There was also no significant reduction after that time [34]. However, pepsin and papain, used in combination at pH 4.8 to 50 °C, also showed a molecular weight decrease. The molecular weight in the initial hydrolysis phase was 500 kDa (91% DDA), and after hydrolysis, it was reduced to 150 kDa after 1 h of reaction. The molecular weight continued to reduce up to 75 kDa in 24 h [22].

When papain (pH 4.5 at 45 °C) was used, the molecular weight decreased rapidly at the beginning of hydrolysis, initially from 350 to 82 and 40 kDa at 30 and 45 min, respectively. From this time on, stability was observed without a significant increase [55]. Otherwise, Celuzyme^®^ XB showed a reduction in molecular weight in the first 135 min and became constant after four hours of reaction due to the decrease in enzymatic activity limiting depolymerization and obtaining glycoside units [31].

Chitosan 371.5 kDa (92% DDA) hydrolyzed by the enzymes lysozyme, chitinase, and cellulase showed a significant decrease in molecular weight in the first three hours hydrolysis to 38, 22, and 15 kDa, respectively. After 20 h, the reduction went to 26, 15, and 7 kDa, respectively. However, there was no significant reduction after that time [53]. Already under the conditions of 50 °C, pH 5, and cellulase to chitosan ratio 1: 5, there was a reduction in the molecular weight of chitosan from 518 kDa to 35, 12, 6, and 1.13 kDa in the times of 1, 4, 8 and 24 h, respectively [33].

### 6.5. Effect of pH on Enzymatic Hydrolysis of Chitosan

The stability and functionality of the enzyme depend on its spatial structure. Chemical changes, such as pH, can modify the structure of an enzyme. The bonds that make up the tertiary structure of the enzyme are altered, which prevents its catalytic function from being efficiently performed. In this way, the pH influences the enzyme activity, where the best pH value or range will be according to the type of enzyme used in hydrolysis. Additionally, chitosan is also influenced by pH because its solubility is low at pH close to 7.0 [31,32].

Under the study conditions with Celuzyme^®^ XB (cellulase/xylanase/β-glucanase) and chitosan, the activity was maximum at pH 4.8. From this pH, the performance of the system tended to decrease gradually. Favorably, the three enzymes are highly stable at acidic pH and, likewise, their efficiency decreases at pH close to neutral [31]. However, in another study using only the cellulase enzyme, the effect was not observed with pH values of 2.0 to 6.0. The enzyme was activated at other pH values, promoting the hydrolysis of chitosan determined by the decrease in viscosity by up to 97% in 1 h [33].

Regarding pepsin, the peak activity in the hydrolysis of chitosan was at pH 4.5. Although the protein pH activity is generally lower, the variability will depend on the specific protein substrate and its native or denatured state [32].

### 6.6. Influence of the Enzyme/Substrate Ratio

In addition to the other parameters already shown, the reaction rate also depends on the enzyme concentration and substrate (chitosan). The reaction rate is better in a higher enzyme/substrate ratio [32,33].

In a constant condition concentration of chitosan, the final viscosity is influenced by the change in enzyme concentration. In this case, the biopolymer degradation increases directly in a higher enzyme concentration. Concentrations of the enzyme Celuzyme^®^ XB greater than 1 × 10^−3^ mg mL^−1^ caused the maximum degradation up to a limit of 2.15 × 10^−3^ mg mL^−1^. After this value, the concentration of reducing sugars is constant [31].

In the study of pepsin, the viscosity of the chitosan solution decreases as the pepsin concentration increases. For hydrolysis times longer than 1 h, an enzyme/substrate ratio of 1: 100 is sufficient to achieve the maximum hydrolysis degree. From this concentration, the reduction in viscosity remains at approximately 80% [32].

A decrease in the hydrolysis rate was observed when the cellulase concentration was kept constant and the chitosan concentration was increased. After 10 min of hydrolysis, the 1: 5 ratio (enzyme: substrate) showed a more significant decrease in viscosity (95%), while the 1:20 ratio was 82% [33]. That is because the viscosity of the system increases with a higher concentration of chitosan. This effect significantly reduces the diffusion of chitosan to the active center of the enzyme. This will decrease enzyme activity [55].

### 6.7. Properties of Chitosan Oligomers

The hydrolytic product of chitosan has more excellent application due to its better solubility and functional characteristics when compared to integral chitosan. The DPPH antioxidant activity was higher in chitosan with a molecular weight below (72 kDa), with a reduction potential of up to 90% with chitooligosaccharide of 2.2 kDa and 38% with medium molecular weight chitosan of 300 kDa. In addition to the molecular weight, it was found that the concentration of chitosan is directly proportional to the antioxidant activity. In chitosan of 2.2 kDa in different concentrations of 1 and 5 mg mL^−1^, the reduction of DPPH was 50% and 65%, respectively [6]. 

In the study with chitosan at a concentration of 1.2 mg mL^−1^ (152 kDa), there was a 15% reduction in DPPH. However, while in chitosan of 64 kDa, the reduction was 75%. Chitosan with higher molecular weight has a compact structure, which promotes the high viscosity of the solution. In these conditions, the strength of the intramolecular hydrogen bond reduces the activity of the hydroxyl and amine groups, limiting the reduction capacity of these groups [31].

Chitosan hydrolyzed with cellulase also had a more significant effect on antimutagenicity. This positive effect was observed at lower molecular weight [6] and evidences a potential use for reducing hypercholesterolemia through binding with bile acids [22]. Chitosan hydrolyzed by a crude enzyme from *Bacillus amyloliquefaciens* V656 also showed antitumor activity after 12 h of hydrolysis. This effect has been related to the presence of hexamer/(GlcNAc)_6_ [10].

Several studies have analyzed the effect of chitosan’s molecular weight on antibacterial activity. Some studies concluded that antibacterial activity increased when there was a reduction in molecular weight. This was observed in 12 kDa low molecular weight chitosan compared to 228 kDa medium molecular weight chitosan against *Bacillus cereus*, *E. coli*, *Staphylococcus aureus*, *Pseudomonas aeruginosa*, *Salmonella* sp, and *Saccharomyces cerevisiae* [58]. It was also observed in 64 kDa chitosan compared to 152 kDa chitosan against *Staphylococcus aureus*, *Pantoea ananatis*, *Pseudomonas aeruginosa*, *Raoultella planticola*, and *Escherichia coli* [31].

Some reports have shown that chitosan of medium molecular weight (194 kDa) has more significant antibacterial activity than chitosan of lower molecular weight (41, 14, and 5 kDa) against Gram-positive microorganisms due to its greater thickness of the peptidoglycan layer, preventing the access of the chitooligosaccharides to the cell. Although the peptidoglycan layer prevents the chitooligosaccharides from accessing the cell, the antimicrobial effect of the higher molecular weight chitosan is due to the formation of a chitosan film. This film can inhibit the absorption of nutrients by the microorganism [34]. Another study, with chitosan from 300 to 3.3 kDa, showed that the pH could influence differently according to the molecular weight of the chitosan, being observed that under acid pH conditions, the chitosan activity increased with the increase of the molecular weight, regardless of the temperature and the bacteria tested. However, at neutral pH, chitosan activity increased as the molecular weight decreased [52].

The antimicrobial activity of low molecular weight chitosan was compared with standard antibiotics. The inhibitory activities of molecular weight between 10 and 100 kDa were as efficient as the antibiotics Flomox^®^ and Kluacid^®^ against *Bacillus cereus*. The specific growth rate of *Candida albicans* was inhabited entirely by chitooligosaccharides of molecular weight less than 1 kDa at a concentration of 0.11 mg mL^−1^ and was more efficient than that of Kluacid^®^ at a concentration of 0.42 mg mL^−1^. These results show that chitosan and low molecular weight chitooligosaccharides can be considered possible alternative antimicrobial agents or additives in pharmaceutical compositions [21].

## 7. Conclusions

From the studies demonstrated, the choice of the type of hydrolysis of chitosan remains a challenge. Several factors influence the process, compromising the yield, cost, and characteristics of the hydrolyzed product. This hydrolyzed product (medium and low molecular weight chitosan and chitooligosaccharides) is responsible for its bioactive properties. New technologies and studies focused on using chitosan hydrolysates have been shown continuously in the scientific literature. However, one point that seems essential is related to the need to develop more efficient hydrolysis technologies to obtain oligomers with specific sizes. Thus, several alternatives of technologies are proposed in isolation or combined to improve process efficiency. The choice of technology to be used depends mainly on the type of application for which the hydrolysis product will be used.

## Figures and Tables

**Figure 1 polymers-13-02466-f001:**
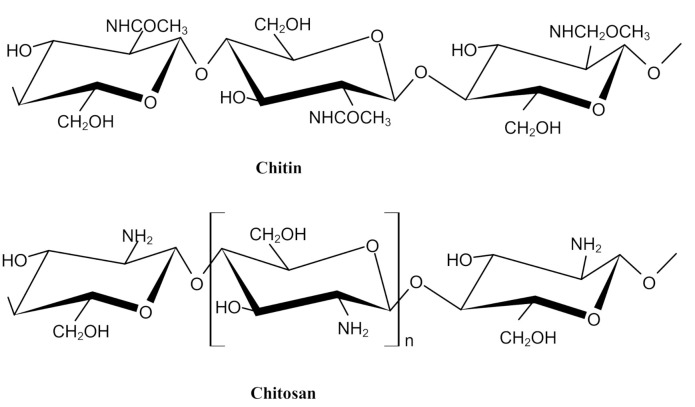
Chemical structure of chitin and chitosan. Reprinted from Vo et al. [1] with permission from John Wiley and Sons and Copyright Clearance Center.

**Figure 2 polymers-13-02466-f002:**
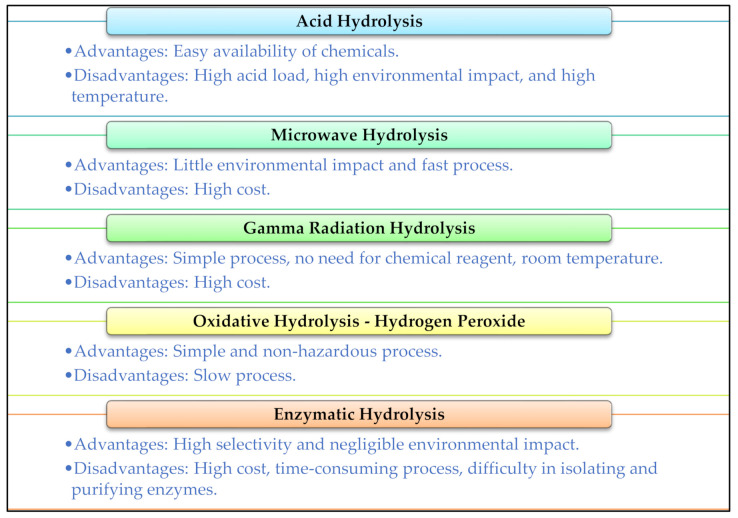
Advantages and disadvantages of the main chitosan hydrolysis methods.

**Figure 3 polymers-13-02466-f003:**
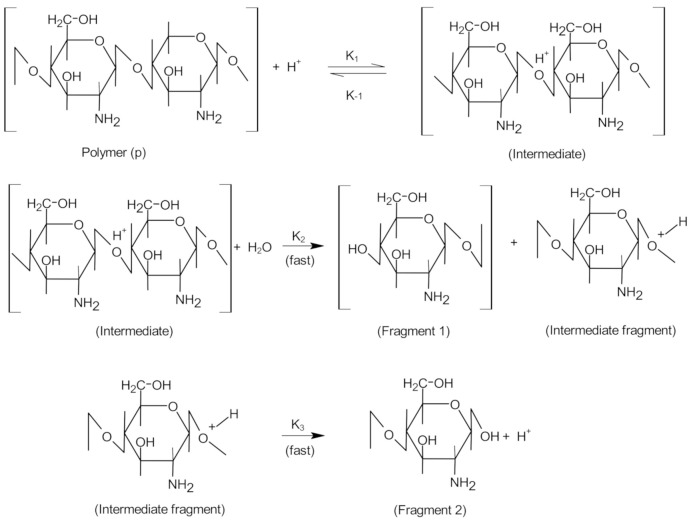
Mechanism proposed for acid hydrolysis of chitosan. Reprinted from Kasaai et al. [24] with permission from Creative Commons Attribution.

**Figure 4 polymers-13-02466-f004:**
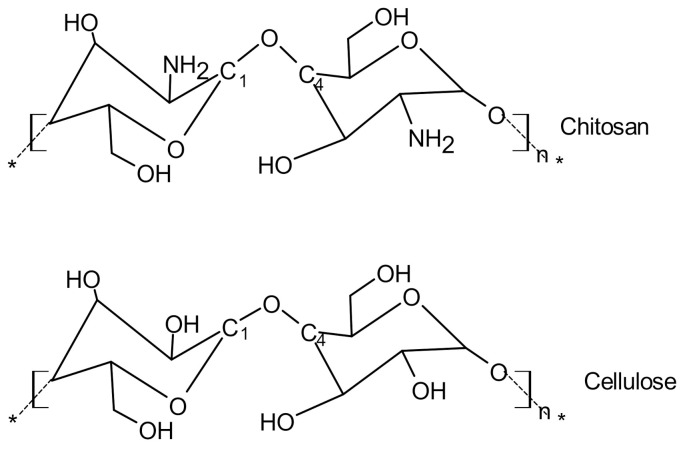
Chemical structure of chitosan and cellulose. Reprinted from Omari et al. [61] with permission Royal Society of Chemistry provided by Copyright Clearance Center.

**Table 1 polymers-13-02466-t001:** Parameters for chitosan acid hydrolysis.

Reference	Chitosan	Solvent	Solvent Concentration	Temperature	Time	Mw/DP
[4]	5%	CH_3_COOH	1–5%	30–60 °C	30–90 min	166.3–592.8 kDa
[9]	3.33%	HCl	6 M	70 °C	2 h	2–12 DP
[15]	0.8%	HCl	0.15 and 1.5 M	25–45 °C	0–60 h	20–90 kDa
[23]	2%	HCl	6.27 M	56 °C	3 h	-
[24]	1%	HCl	0.1–5 M	65 °C	13–2160 min	73.8–1076 kDa
[25]	0.5%	HCl	8, 10, and 12 M	90 and 105 °C	0–10 h	-
[26]	255 mg	([C4mim] Cl ^1^ or [C4mim]Br ^2^) + (H_2_SO_4_ or HCl or HNO_3_) + (Water)	(4g) + (290–600 mg)+ (27–108 mg)	100 °C	190–540 min	-
[27]	1%	HCl	2 M	-	0.5–24 h	1.5–29 kDa
[35]	1%	C_3_H_6_O_3_	1%	8, 22, and 37 °C	10, 20, 30, 60, and 90 days	-
[36]	5%	H_3_PO_4_	85%	Ambient; (40, 60, and 80 °C)	35 days; (1–15 h)	19–164 kDa
[37]	0.2%	HCl:H_3_PO_4_	100:0, 75:25, 50:50, 25:75, 0:100 (6M H^+^) 75:25 (4, 6, 8, and 10 M H^+^)	110 °C	0–36 h	-

^1^ [C4mim]Cl: 1-butyl-3-methylimidazolium chloride; ^2^ [C4mim]Br: 1-butyl-3-methylimidazolium bromide.

**Table 2 polymers-13-02466-t002:** Parameters of chitosan hydrolysis by gamma radiation.

Reference	Chitosan	Solvent	Solvent Concentration	Radiation Doses (kGy)	Rate (kGy/h)	Mw (kDa)
[7]	-	(NH_4_)_2_S_2_O_8_ or K_2_S_2_O_8_ or H_2_O_2_	10%	20 to 200	6.7	130–3000
[8]	dry/wet/solution	wet (H_2_O)/solution (CH_3_COOH/H_2_O_2_)	1%	15 to150	1.02	10–180
[11]	2%	CH_3_COOH	1%	25 to 100	-	82.2–337.73
[20]	1:6 (chitosan:water)	H_2_O_2_	1–5%	6	-	8–14
[28]	2%	CH_3_COOH	1%	2, 10 and 20	3.35	2.1–35.2
[29]	5%	CH_3_COOH/H_2_O_2_	0.2 M/1%	0 to 150	3	6–100
[40]	5%	C_3_H_6_O_3_/H_2_O_2_	3%/1%	4 to 16	1.33	2.7–8.6
[41]	2%	CH_3_COOH	2%	2 to 200	-	-
[42]	20%	H_2_O_2_	2%, 10%, and 30%	10 to 100	10	1–300
[43]	0.1–2%	CH_3_COOH	0.1 M	0.5 to 200	10	0.97–67

**Table 3 polymers-13-02466-t003:** Parameters of chitosan hydrolysis by microwave.

Reference	Chitosan	Solvent	Solvent Concentration	Other Solutions	Power	Time	Temperature	Mw (kDa)
[17]	3%	CH_3_COOH; H_2_O_2_	2%, 1%	-	800 W	25 min	80 °C	1.46
[18]	3%	AmimCl ^a^: HmimCl ^b^	(9:1)	15 mg (H_2_O) + DMSO ^c^ e SFILs ^d^	160, 320, 640, 800 W	30, 60, 90, 120 s	120, 125, 130, 146, and 155 °C	0.45–24
[19]	1%	HCl	3 M	-	80–100 W	5, 10, and 15 min	80 and 100 °C	-
[30]	1%	CH_3_COOH	0.1 M	-	10–100 W	5–80 min	-	25–42
[45]	3 e 4%	H_2_O_2_	5, 10 e 15%	-	700 W	3, 4, and 5 min	-	0.9–1
[46]	-	acidified	-	-	650 or 390 W	10, 20, 30, and 60 min	98–100 °C	79.2–142.2
[47]	-	CH_3_COOH or HCl	2%	NaCl, KCl, CaCl_2_ (0.15 mol/L [Cl^−1^]	480–800 W	0.5–25 min	100 °C	30–105

^a^ AmimCl: 1-Ally-3-methylimidazolium chloride; ^b^ HmimCl: 1-H-3-methylimidazolium chloride.; ^c^ DMSO: Dimethyl Sulfoxide.; ^d^ SFILs: Sulfonic acid-functionalized ionic liquids.

**Table 4 polymers-13-02466-t004:** Parameters of chitosan hydrolysis by hydrogen peroxide.

Reference	Chitosan	Solvent	Solvent Concentration	Temperature	Time	Mw/DP
[12]	1%	H_2_O_2_; CH_3_COOH	0.5–3%; 1%	50–75 °C	1–6 h	2–7 DP
[16]	1%	H_2_O_2_; CH_3_COOH; H_3_PW_12_O_40_	0.5–3%; 1%; 0.04–0.14%	50–75 °C	10–60 min	7 DP
[17]	3%	H_2_O_2_; CH_3_COOH	3%; 2%	80 °C	180 min	1.36 kDa
[46]	1%	H_2_O_2_; CH_3_COOH	5.7, 2.8, 1.9, and 1.7 (molar ratio)	50 °C	8 h	6.61–9.97 kDa
[48]	2%	H_2_O_2_; CH_3_COOH	2%; 1%	40 °C	30–180 min	-
[49]	2%	H_2_O_2_; HCl	(0.5, 1.0, 1.5, 2.0 M)/0.5%	25, 40, 50, 70 °C	1, 2, 3 h	11–1200 kDa
[50]	7.5%	H_2_O_2_; H_3_PW_12_O_40_	4.5%; 0.1%	70 °C	30–120 min	4.3–4.7 kDa
[51]	-	H_2_O_2_; HCl	0–5%; 0–9%	10–90 °C	0.5–8 h	5–200 kDa

**Table 5 polymers-13-02466-t005:** Parameters of chitosan enzymatic hydrolysis.

Reference	Enzime	Enzyme Concentration ^1^	Chitosan ^2^	Solution	pH	Temperature	Time	Mw/DP
[6]	Cellulase	10 U/g of chitosan	1%	Sodium acetate 0.5 M	5.2	55 °C	1–24 h	2.2–156 kDa
[10]	Not identified	20% ^2^	1%	Phosphate 0.05 M	5.0	37 °C	1–24 h	1–6 DP
[21]	Chitosanase	0.95 U/mg		Sodium acetate	5.6	55 °C	1.5 h	1–100 kDa
[22]	Pepsin + papain	4%	1%	Sodium acetate 0.2 M	4.8	50 °C	0–24 h	0.6–150 kDa
[31]	Celuzyme^®^ XB (cellulase/xylanase/β-glucanase)	0–2.5 × 10^−3^ mg/mL	0.5%	Acetic acid-sodium acetate 0.2 M	3.5–6.5	25–75 °C	25–250 min	64–152 kDa
[32]	Chitosanase, cellulase, hemicellulase, papain, bromelain, pepsin, protease type XIV, lysozyme, and lipase A.	0.1–10%	1%	Sodium acetate 0.1 M	3.0–5.0	30–50 °C	0–20 h	-
[33]	Cellulase	5–20%	0.5–4%	Acetic acid	2.0–6.0	40–60 °C	5 min–24 h	85.8–1.13 kDa
[46]	Pectinase + cellulase + papain	15+15+2%	2%	Phosphate 1 M	5.3	39 °C	24 h	14.6–34 kDa
[52]	Cellulase	10 U/g of chitosan	-	Acetate-bicarbonate 0.5 M	5.2	55 °C	1–18 h	3.3–156 kDa
[34]	Papain/lysozyme/cellulase	0.003%	1%	Sodium acetate 0.1 M	4.0	40/30/37 °C	1–16 h	4.3–800 kDa
[53]	Chitinase, cellulase, or lysozyme	10% ^2^	2.2%	Sodium acetate 0.1 M	4.0	42 °C	0–180 h	6–38 kDa
[54]	Cellulase	0.5 mL	1%	Sodium acetate 0.02 M	5.2	60 °C	0.5–12 h	2–4 DP
[55]	Papain	0.08–0.12 g/g of chitosan	6–10 g/L	Sodium acetate 0.2 M	4.0–5.0	40–50 °C	15–120 min	35–155 kDa
[56]	Chitosanase	1.5 U/mL	1%	Sodium acetate 0.02 M	5.5	30 °C	0.5–6 h	2–7 DP
[57]	Not identified	0.05 U/mL	1%	Sodium acetate 0.05 M	5.0	60 ºC	-	2–5 DP
[58]	Cellulase	10 U/g of chitosan	4.5%	Acetic acid-bicarbonate 0.5 M	5.2	55 °C	18 h	12 kDa/2–8 DP
[59]	Cellulase	20%	5%	Acetic acid 1%	5.6	50 °C	24 h	3–11 DP

^1^ Regarding the substrate; ^2^ Regarding the solution.

## Data Availability

The data presented in this study are available on request from the corresponding author.

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
