# Peer review of "Production of Low Molecular Weight Chitosan and Chitooligosaccharides (COS): A Review"

_polymers, 2021, doi:10.3390/polym13152466_

Round 1

Reviewer 1 Report

This paper shows a good review of Low Molecular Weight Chitosan and Chitooligosaccharides. There are some issues that need to address:

- Introduction is written simply, most recent research and innovation in Chitooligosaccharides performances should be reviewed to show the gap of knowledge. The introduction should be extended with recently research papers.

what makes this review different from the others and from the most recent ones?

- section of drawbacks and future could be increased quality of the manuscript.

There are some grammatical errors, please carefully check the whole manuscript.

- Abstract should be rewritten. The general information should be concisely. Instead, more details of reviewed aspects should be presented.

- A review paper not only should summarize recently published works, but also should contain critical and comprehensive discussions. Therefore, check writing for the whole manuscript. The review should not be presented by listing what has done by others.
- Technical terms are misused through the manuscript and the writing needs a revision.

Author Response

We thank the reviewers for taking the time and careful reading and evaluation of this manuscript.  We appreciated the suggestions, and we are confident that the manuscript has been enhanced with the changes.

Our responses to the recommendations are listed below (indicated in the manuscript with blue color).

Best regards.

  1. Introduction is written simply, most recent research and innovation in Chitooligosaccharides performances should be reviewed to show the gap of knowledge. The introduction should be extended with recently research papers

Response:  The introduction has been modified according to the reviewer's suggestions (pages 1-4). Thank you.

  1. “Section of drawbacks and future could be increased quality of the manuscript.”

Response: Thank you for this suggestion. Was added a figure describing the advantages and disadvantages of using chitosan hydrolysis methods (page 3).

  1. “Abstract should be rewritten. The general information should be concisely. Instead, more details of reviewed aspects should be presented.”

Response: The abstract was modified according to the suggestions given by the reviewer (page 1). Thank you.

  1. “Technical terms are misused through the manuscript and the writing needs a revision.”

Response: Sorry for the errors. They were removed and modified throughout the text.

Reviewer 2 Report

This review paper compares different hydrolysis methods (acid hydrolysis, oxidative hydrolysis, enzymatic hydrolysis, gamma radiation-induced hydrolysis and microwave-enhanced hydrolysis) used for production of low molecular weight chitosan and chitooligosaccharides. The influence of different parameters of the reaction process on the resulting properties of chitosan oligomers obtained is considered in details. The data summarized in this review may be useful for both organic chemists working on the improvement of chitosan hydrolytic methods and the wide range of specialists dealing with biomedical applications of chitosan derivatives, since the brief comparison of the main properties of the available chitooligosaccharides can help them to choose the most suitable product. There are no significant mistakes throughout the text, so the paper is recommended to be published in Polymers in its present form without any additional revisions.  

Author Response

We thank the reviewers for taking the time and careful reading and evaluation of this manuscript.  We appreciated the suggestions, and we are confident that the manuscript has been enhanced with the changes.

Our responses to the recommendations are listed below (indicated in the manuscript with blue color).

Best regards.

  1. This review paper compares different hydrolysis methods (acid hydrolysis, oxidative hydrolysis, enzymatic hydrolysis, gamma radiation-induced hydrolysis and microwave-enhanced hydrolysis) used for production of low molecular weight chitosan and chitooligosaccharides. The influence of different parameters of the reaction process on the resulting properties of chitosan oligomers obtained is considered in details. The data summarized in this review may be useful for both organic chemists working on the improvement of chitosan hydrolytic methods and the wide range of specialists dealing with biomedical applications of chitosan derivatives, since the brief comparison of the main properties of the available chitooligosaccharides can help them to choose the most suitable product. There are no significant mistakes throughout the text, so the paper is recommended to be published in Polymers in its present form without any additional revisions.”

Response: We thank the reviewer for the availability, evaluation, and recognition of this manuscript. The authors believe that the article can contribute to different areas of research.

Round 2

Reviewer 1 Report

The authors considered the comments of the reviewer. The revised manuscript is improved. However, It is recommended to prepare statistical data (such as the number of documents, document per country) about you used references by created databank such as Scopus, Google scholar, and web of science (for introduction part).

Author Response

Title: “Production of Low Molecular Weight Chitosan and Chitooligosaccharides (COS): A Review”

 We thank the reviewers for taking the time and careful reading and evaluation of this manuscript.  We appreciated the suggestions, and we are confident that the manuscript has been enhanced with the changes.

Our responses to the recommendations are listed below (indicated in the manuscript with blue color).

Best regards.

The authors considered the comments of the reviewer. The revised manuscript is improved. However, It is recommended to prepare statistical data (such as the number of documents, document per country) about you used references by created databank such as Scopus, Google scholar, and web of science (for introduction part).

Response: We appreciate your suggestion. We have added text in the introduction (lines 121-123) and supplementation in appendix A.
